# Diversity of Fish and Decapod Fry in the Coastal Zone of Amvrakikos Gulf

George Katselis [1] , Nikolaos Vlahos [1] , Constandin Koutsikopoulos [2] and Dimitrios K. Moutopoulos [1],*

[1] Department of Fisheries & Aquaculture, University of Patras, 30200 Mesolonghi, Greece; gkatselis@upatras.gr (G.K.); vlachosn@upatras.gr (N.V.)
[2] Department of Biology, University of Patras, 26504 Rio-Patras, Greece; ckoutsi@upatras.gr
* Correspondence: dmoutopo@upatras.gr

**Abstract:** Amvrakikos Gulf and its surrounding coastal lagoons are of primary importance for the local biodiversity and fishing activities. Fish species inhabited the coastal lagoons based on the seasonal ongoing migration movements of fry and adult fish individuals from the sea towards the lagoons. Information on the early stages of fish and decapod species in the Amvrakikos Gulf is limited only to the planktonic ontogenetic stages and reproduction biology, respectively. The aim of this study was to describe the spatial distribution of fry from commercially important fish and decapod species in the coastal zone of Amvrakikos Gulf. The seasonal appearance of the early stage of the most commercially important fish species caught in the coastal zone of the gulf ranged from one to four seasons, depending on the species. Individuals of all ontogenetic stages (fry, juveniles, and adults) were reported for several species (*A. boyeri*, A. fasciatus, *S. abaster*, *S. tyfle*, and *B. ocellaris*), indicating that these species may be regarded as residents in the coastal zone, providing habitats for their entire life cycle. The average relative abundance of the species/genera exhibited no differences compared to other Greek brackish waters. The species composition in the Amvrakikos Gulf at 10 cm and above was in agreement with the transitional nature of the area, with permanent and occasional species present. The present study emphasizes the importance of the coastal zone as a nursery habitat for commercially important species.

**Keywords:** fish migration movements; coastal lagoons; biodiversity; Greece

## 1. Introduction

Lagoons constitute a crucial component of the coastal environment that is influenced by both the marine and terrestrial environments due to their location between land and sea, exhibiting (e.g., [1,2]) (a) food and habitat for numerous fish species, (b) variable abiotic conditions (i.e., salinity, temperature, dissolved oxygen, and type of the substrate), (c) great sensitivity to climate changes, and (d) vulnerability to human disturbance due to their confinement from the open sea and their shallowness. Around the Mediterranean, a coastal area of more than 6500 km$^2$ is covered by coastal lagoons that are exploited by local fisher communities [3]. Lagoons comprise distinct aquatic ecosystems that are either covered by the Natura 2000 network (http://ec.europa.eu/environment/nature/natura2000/index_en.htm, accessed on 20 October 2023) or are protected under the Ramsar treaty (www.ramsar.org, accessed on 20 October 2023).

Amvrakikos Gulf (Western Greece, Ionian Sea) (Figure 1) is a fjord-like hydrological regime [4] (400 km$^2$), protected by national and international regulations for its diverse wildlife and wetlands [5]. The coastal zone up to 5 m in depth covers 88.3 km$^2$ (≈22% of the gulf area) (Figure 1). There are 14 lagoons surrounding the gulf, totaling about 86 km$^2$ (about 20% of the surface area of Greek lagoons). Two rivers—the Arachthos and Louros rivers—drain into the gulf from the gulf's north and northwest coast. Only small-scale fisheries are allowed to operate in the Gulf, as since 1953, trawling and purse-seining

have been forbidden all year round (Royal Fishing Law No23.3/8-4-53). The professional fishing fleet consists of vessels that only fish within the Gulf, primarily pursuing red mullet (*Mullus barbatus* (Linnaeus, 1758)), common cuttlefish (*Sepia officinalis* (Linnaeus, 1758)), ceramote prawn (*Penaeus kerathurus* (Forsskål, 1775)), sand steenbras (*Lithognathus mormyrus* (Linnaeus, 1758)), European pilchard (*Sardina pilchardus* (Walbaum, 1792)), Mugilidae, and *Solea* spp. [6]. Fisheries have also been linked to the preservation of the lagoon environment and the well-being of local communities [5]. Fishery exploitation in the study area, as in the majority of Greek lagoons, is based on seasonal ongoing migration movements of fry and adult fish individuals between sea and lagoons [5,7]. Fish are allowed to enter the lagoons from February to May for feeding and shelter, after which the entrapment devices are open, allowing the free movement of the fish.

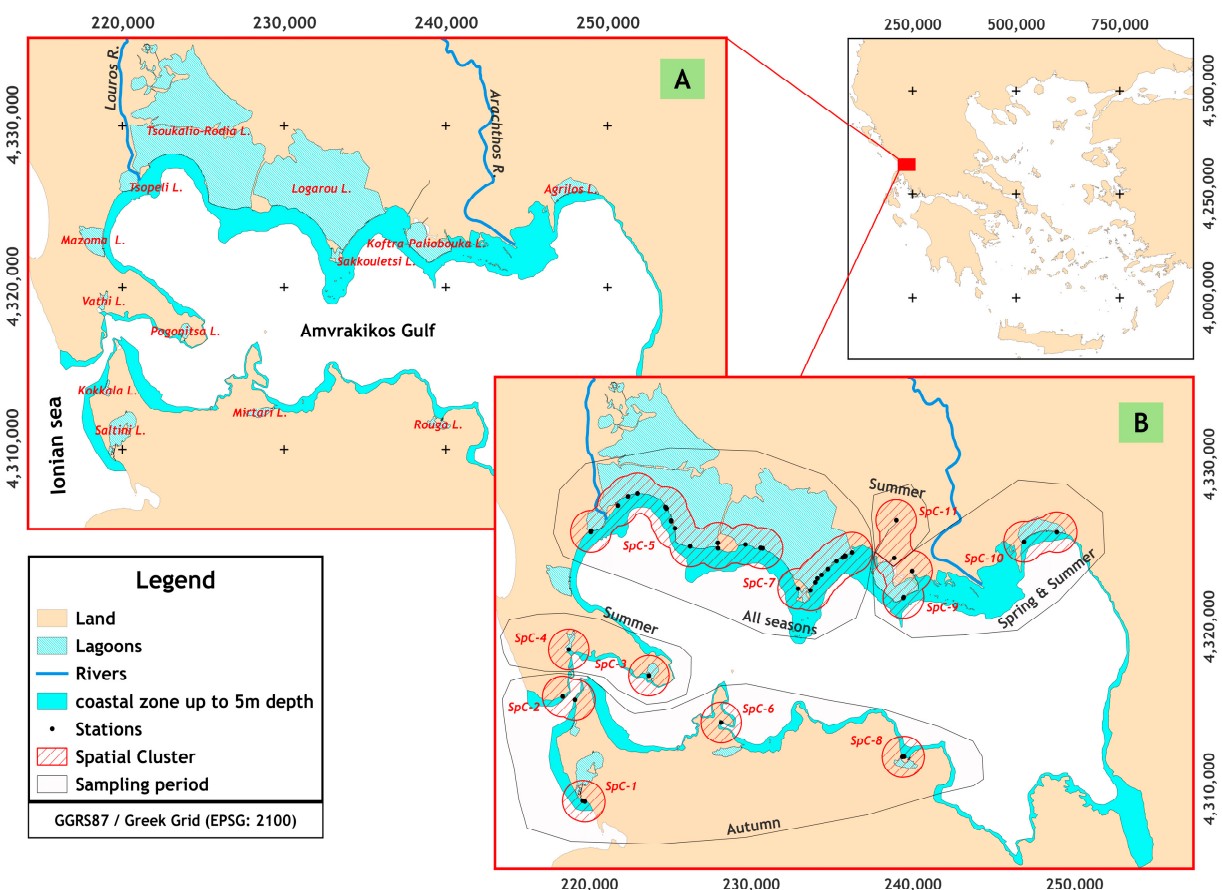

**Figure 1.** Map of the study area (**A**) and sampling station for early stages species in the coastal zone of Amvakikos Gulf, May 2016 to March 2017, station's spatial clusters (SpC-i), and sampling period (**B**).

Like other Mediterranean lagoons (e.g., Albania, Tunisia, and Egypt [3]), the majority of the Amvrakikos' lagoons are in the public domain, leasing to fisher's associations or private fishers. Fisheries exploitation is conducted using permanent traps established at the interface between the lagoon and open sea following species-specific inshore–offshore seasonal movements [5], whereas small-scale fishing gears such as trammel and fyke nets are used complementary. Mugilidae, i.e., flathead grey mullet (*Mugil cephalus* (Linnaeus, 1758)), thinlip grey mullet (*Chelon ramada* (Risso, 1827)), leaping mullet (*Chelon saliens* (Risso, 1810)), golden grey mullet (*Chelon auratus* (Risso, 1810)), thicklip grey mullet (*Chelon labrosus* (Risso, 1827)), gilthead seabream (*Sparus aurata* (Linnaeus, 1758)) and eels (*Anguilla anguilla* (Linnaeus, 1758)) are the most representative species caught contributing more than 80% of total lagoon landings [7]. To a lesser extent, other species also caught were gobies (*Gobius* sp.), soles (*Solea* spp.), sea bass (*Dicentrarchus labrax* (Linnaeus, 1758)), *P. kerathurus*, and Sparidae (annular sea breams, *Diplodus annularis* (Linnaeus, 1758)), striped

sea breams (*Diplodus sargus* (Linnaeus, 1758)), two banded sea breams (*Diplodus vulgaris* (Geoffroy Saint-Hilaire, 1817)), big-scale sand smelt (*Atherina boyeri* (Risso,1810)), and sharp snout sea breams (*Diplodus puntazzo* (Walbaum, 1792)) [7].

The preservation and management of lagoon fish populations increasingly depend on the preservation of nursery habitats, such as the lagoons, for commercially significant species, as well as continuous fishing monitoring. However, the lack of crucial information on each species' ecology (e.g., the amount of recruitment and the standards that young fish in the lagoon basin adopt when choosing their habitat) makes proper management difficult [8]. Studies on the early stages of fish and decapod species in the above-mentioned protected area are limited only to the ontogenetic stages of the planktonic fish [9] and the reproduction biology of *P. kerathurus* [10], respectively. The spawning habitats for most of the above-mentioned demersal species are situated in the open sea (Ionian Sea) [9], for eel in the Sargasso Sea [11], and for *P. kerathurus* inside the Gulf [10]. The lack of crucial information on the ecology of species, such as habitat preferences for young fish, is mentioned as a challenge. During 1980–2005, an important reduction of mullets and an increase in sea bream annual landings were recorded mainly due to possible changes in fry abundance in the coastal zone in proximity to the lagoons [7]. Thus, the aim of this study was to describe the spatial distribution of fry from commercially important fish and decapod species in the coastal zone of Amvrakikos Gulf. The seasonal appearance and the relative abundance of the early stages of the study fish species were also compared with the results from similar studies in other Greek brackish waters and raised the concept of transitional zones between lagoons and coastal marine habitats.

## 2. Materials and Methods

To determine the presence and abundance per species, monthly samplings at 79 stations were carried out from May 2016 to March 2017 in the northern part of the Amvrakikos Gulf (Figure 1) using a seine net of 10 m length, 1.2 m in width and 2.0 mm of the mesh size. The samples were preserved in 4% formalin, and the total length was measured in the laboratory. Relative abundance was estimated by the catch per unit effort method (CPUE), which was defined by the area covered by seine net hauling and converted to 100 m$^2$. New recruits (NR) are defined as individuals with a total length smaller than 2 or 3.5 cm based on the biological characteristics of each species (Froese and Pauly 2023). In each station, the substrate type, plant coverage, depth, water salinity, and temperature were recorded. The stations were grouped into clusters (SpC$_i$) based on the distance among the stations, lower or equal to 2.5 km. Furthermore, a disaggregation takes place using the salinity gradient as a criterion (Figure 1: SpC-9 (S = 23–26 PSU) and SpC-11 (S = 0.5 PSU)).

To identify similarities in the microhabitat, a clustering technique based on Ward's method and squared Euclidean distance was applied to abiotic parameters (substrate type, plant coverage, and depth) among stations. For the cluster analysis, the muddy, muddy sand, sand, and sandy gravel substrates received the values 1, 2, 3, and 4, respectively. The Shannon–Wiener diversity index (H) [12] was used to measure the diversity between the species composition among the SpC, microhabitat clusters (SC), and season (January–March: 1; April–June: 2; July–September: 3; and October–December: 4). To test for differences of species relative abundance of NR and all stages (AS) among the SpC, SC, and season (Kruskal–Wallis test and for post hoc Mann–Whitney U test), was used [13]. The uni-variate chi-square ($\chi^2$) procedure was performed to examine the possible relation in the ratios between groups of spatial clusters. All statistical analyses were carried out using the SPSS package ver. 28.0.1.0.

In order to ascertain each species' ecological profile, species were categorized based on their frequency of occurrence into two SpC groups: high spatial distribution, which was defined as species occurring in more than 50% of the SpCs samplings (score A = 2), and low spatial distribution, which was defined as species occurring in equal to or less than 50% of the SpCs samplings (score A = 1). In order to ascertain species in terms of seasonality, species were categorized as non-seasonal when they occurred in more than

50% of the number of seasons (score B = 2) and as seasonal when they occurred in equal to or lower than 50% (score B = 1) of the number of seasons, respectively. In order to ascertain species in terms of microhabitat clusters (SC), species were categorized as non-selective in microhabitats when the species occurred in more than 50% of SCs (score C = 2) and selective in microhabitats when the species occurred in equal to or lower than 50% (score C = 1) of the number of SC, respectively. The overall score for AS and NR of each species was estimated as follows:

$$\text{total score} = 100 * \text{score A} + 10 * \text{score B} + \text{score C}$$

Thus, a species was highly spatially distributed, non-seasonal, and non-selective in the microhabitat when the total score was 222, highly spatially distributed, non-seasonally, and non-selective in the microhabitat when the total score was 221, highly spatially distributed, seasonal, and non-selective in the microhabitat when the total score was 212, and highly spatially distributed, non-seasonal, and selective in microhabitat when the total score was 211.

## 3. Results

Overall, 11,880 specimens were caught belonging to twenty-one families (eighteen families of fish, two decapods, and one cephalopod) and twenty-three species/genera of fish, three species/genera of decapods, and one cephalopod (*Sepia officinalis*). Total length ranged from 0.60 to 21.20 cm, with a mean value of 4.18 (SD: 2.26 cm) (Table S1 in Supplementary Materials). Fish represented 94.8% (11,259 individuals) of the total specimens caught, with the most representative family being Atherinidae (35.6%), followed to a lesser extent by Gobiidae (14.8%), Cyprinodontidae (13.5%), Mugilidae (12.7%), and Syngnathidae (8.2%), whereas the remaining families cumulatively represented 7.3% of total number of specimens (Table S1 in Supplementary Materials). Half of the specimens sampled included *A. boyeri*, *D. labrax*, *C. saliens*, *C. aurata*, *C. ramada*, *M. cephalus*, *S. solea*, *S. aurata*, and *P. kerathurus*, followed to a lesser extent (20.2%) by the commercial interest species *Sardina pilchardus* and *Sepia officinalis*, protected species (i.e., *Aphanius fasciatus*, *Telestes pleurobipunctatus*, and *Singathus abaster*), and one invasive species (*Gambusia holbrooki*) (2.5%) (Table S1 in Supplementary Materials).

More than 50% of the individuals of *D. labrax* (100%), *M. cephalus* (97.0%), *S. aurata* (94.7%), *Palaemon* sp. (91.0%), *P. kerathurus* (77.9%), *C. aurata* (74.3%), *C. saliens* (60.7%), and *G. holbrooki* (55.5%) were NR. On the remaining species/genera, the NR ranged from 1.2% (*Belone belone*) to 44.4% (shrimp) of their sampled individuals (Table S1 in Supplementary Materials). The overall relative (mean $\pm$ SD) abundance for AS was 137.3 $\pm$ 103.6 ind. 100 m$^{-2}$, while the most abundant species/genera were *G. holbrooki*, *A. boyeri*, *E. encrasicolus*, *A. fasciatus*, and *C. saliens*. For the remaining species/genera, the relative abundance for AS ranged from 0.78 $\pm$ 0.38 (*Hippocampus* sp.) to 28.29 $\pm$ 62.75 ind. 100 m$^{-2}$ (*Gobius* sp.). In contrast, the overall relative abundance for NR was 43.47 $\pm$ 44.44 ind. 100 m$^{-2}$, while the more abundant species/genera were *G. holbrooki*, *C. saliens*, *S. aurata*, and *A. boyeri*. For the remaining species/genera, the relative abundance for NR ranged from 0.87 $\pm$ 0.29 (*Syngnathus typhle*) to 12.42 $\pm$ 18.15 ind. 100 m$^{-2}$ (*A. fasciatus*) (Table S1 in Supplementary Materials).

The overall H$'$ were 2.14 and 2.15 for AS and NR, respectively, while the mean $\pm$ SD H$'$ per station were 1.01 $\pm$ 0.47 (min–max: 0–2.09) and 0.69 $\pm$ 0.47 (min–max: 0–1.82) for AS and NR, respectively. The number of species in the overall sample was twenty-five and twenty species for AS and NR, respectively, while per each station, the mean $\pm$ SD number of species was 5.79 $\pm$ 2.65 (min–max: 1–13 species) and 3.29 $\pm$ 1.86 (min–max: 1–9 species) for AS and NR, respectively (Table S1 in Supplementary Materials). Regarding the AS for SpC, the number of species present ranged from two (SpC-4) to twenty-one (SpC-5), while for each season, it ranged from fifteen (season 1) to nineteen (seasons 2 and 4), and for each SC from fifteen (SC4) to twenty-three (SC2). The presence of each species ranged from one

SpC (*D. labrax* (SpC-5), *G. holbrooki* (SpC-11), *S. scrofa* (SpC-1), *T. pleurobipunctatus* (SpC-11)) to ten SpCs (*A. boyeri* (absent in SpC-11)) (Table S2, Figure S1 in Supplementary Materials).

Cluster analysis applied among station physicochemical features identified four groups (SCi) (Table 1). Cluster SC1 was characterized by the sandy substrate, high plant coverage, low depth, and lowest salinity. Cluster SC2 was characterized by the sandy gravel substrate, high plant coverage, medium depth, and higher salinity. Cluster SC3 was characterized by the muddy sand substrate, low plant coverage, low depth, and low salinity, while Cluster SC3 by the muddy substrate, low plant coverage, high depth, and low salinity (Kruskal–Wallis test: >26, $p < 0.05$; post hoc Mann–Whitney U test, $p < 0.05$). At SC1, SC2, SC3, and SC4, 13, 29, 15, and 22 stations were grouped, respectively (Table 1).

**Table 1.** Results of cluster analysis on the substrate type, plant coverage (%), depth of stations, and results of statistical analysis of abiotic features per cluster. The letter index marks the station cluster, which differs statistically from others per abiotic feature (column). The same letter means non-statistically significant differences (Kruskal–Wallis test and for post hoc Mann–Whitney U test).

| Station Cluster | Substrate Type * | Mean Value (Standard Deviation) | | | | Station/SpC | | Number |
| | | Plant Coverage % | Depth (m) | Tem (°C) | Sal (PSU) | SpC-Others | SpC-5 and SpC-7 | Total |
|---|---|---|---|---|---|---|---|---|
| SC1 | 3.00 (0.0) [c] | 61.53 (38.69) [b] | 0.47 (0.15) [a] | 21.9 (4.51) | 25.5 (6.94) [a,b] | 3 | 10 | 13 |
| SC2 | 3.96 (0.18) [d] | 55.51 (30.88) [b] | 0.66 (0.26) [b] | 21.0 (5.68) | 29.4 (3.77) [b] | 8 | 21 | 29 |
| SC3 | 2.00 (0.00) [b] | 9.33 (10.32) [a] | 0.32 (0.07) [a] | 19.5 (7.02) | 22.3 (4.86) [a] | 1 | 14 | 15 |
| SC4 | 1.04 (0.21) [a] | 15.45 (13.70) [a] | 1.44 (0.36) [c] | 21.9 (6.38) | 22.1 (6.83) [a] | 5 | 17 | 22 |
| Total | 2.62 (1.22) | 36.58 (33.88) | 0.78 (0.49) | 21.1 (5.94) | 25.4 (7.52) | 17 | 62 | 79 |
| Range | 1–4 | 0–100 | 0.3–1.7 | 12–36 | 0.5–48 | | | |
| K–W Test | 76.45 | 34.31 | 54.47 | 2.22 | 26.07 | 2.62 ($\chi^2$; df = 3) | | |
| Sig. | <0.05 | <0.05 | <0.05 | >0.05 | <0.05 | <0.05 | | |

* 1: mud; 2: muddy sand; 3: sand; 4: sandy gravel.

The sampling stations were grouped in 10 spatial clusters (SpC-j) (Figure 1). At pooled groups of spatial clusters, SpC-5, SpC-7, and SpC-others consisted of 62 and 17 stations, respectively. The composition of SCi did not differ ($\chi^2 = 2.62$; df = 3; $p > 0.05$) between the two pooled groups of spatial clusters SpC-5 and SpC-7 and SpC-others (Table 1). The groups SpC-1 and SpC-11 consisted of stations located at marine and freshwater regimes, respectively.

The seasonal presence of each species ranged from one season (*D. labrax* (season 1), *G. holbrooki* (season 4), *S. scrofa* (season 2), *T. pleurobipunctatus* (season 4), and *M. cephalus* (season 4)) to four seasons (nine species/genera). According to SC, the seasonal presence of each species ranged from one SC (*G. holbrooki* (SC4), *S. scrofa* (SC2)) to four SCs (fourteen species/genera) (Table S2 in Supplementary Materials). Significant (Kruskal–Wallis test; post hoc Mann–Whitney U test; $p < 0.1$) higher relative abundance per station was shown for five species in SpC (*A. fasciatus* (SpC-5 and SpC-8), *A. boyeri* (SpC-3, SpC-5, and SpC-7), *C. aurata* (SpC-1, SpC-2, and SpC-5), *E. encrasicolus* (SpC-5 and SpC-7), and *M. cephalus* (SpC-1)), five species/genera in season (*A. boyeri* (season 3), butterfly blenny *(Blennius ocellaris (Linnaeus, 1758))* (seasons 2 and 3), *C. ramada* (season 1), shrimp (seasons 3 and 4), and *S. abaster* (season 2)), and six species/genera in SC (*A. fasciatus* (SC3), *B. ocellaris* (SC1 and SC2), *Palaemon* sp. (SC3 and SC4), shrimp (SC3 and SC4), and *S. abaster* and *S. typhle* (SC1)) (Table S2 in Supplementary Materials).

Regarding the NR for SpC, the number of species presented ranged from one (SpC-4) to eighteen (SpC-5), and for SC, from thirteen (SC4) to nineteen (SC2). The seasonal presence of each species/genus ranged from no presence for *E. encrasicolus*, *Hippocampus* sp., *L. viridis*, *S. scrofa*, and *T. pleurobipunctatus* to four seasons for *A. boyeri*, *Gobius* sp., and *Palaemon* sp. According to SC, the seasonal presence of each species/genus ranged from no presence for *E. encrasicolus*, *Hippocampus* sp., *L. viridis*, *S. scrofa*, and *T. pleurobipunctatus* to four SCs for *A. fasciatus*, *A. boyeri*, *C. aurata*, *C. saliens*, *Palaemon* sp., *P. kerathurus*, *S. aurata*, and shrimps (Table S2 in Supplementary Materials).

Significant (Kruskal–Wallis test; post hoc Mann–Whitney U test; *p* < 0.1) higher relative abundance per station was shown for two species/genera in SpC (*C. aurata* (SpC-2) and *M. cephalus* (SpC-1)), seven species/genera in season (*A. boyeri* (seasons 2 and 3), *C. aurata* (season 4), C. *saliens* (season 3), *P. kerathurus* (seasons 2 and 3), *S. aurata* (Season 1), shrimp (season 1), and S. abaster (season 2)), and five species/genera in SC (*A. fasciatus* (SC3), *B. ocellaris* (SC2), *Palaemon* sp. (SC3 and SC4), shrimp (SC3), and *S. abaster* (SC1 and SC4)) (Table S2 in Supplementary Materials). The H′ and S per station showed non-significant differences (Kruskal–Wallis test, *p* > 0.1) among the SpCs for AS and NR. In contrast, the H′ and S were significantly different (Kruskal–Wallis test, *p* < 0.1) among season and SCs stations (Figure 2).

## Shannon-Wiener Index (H')

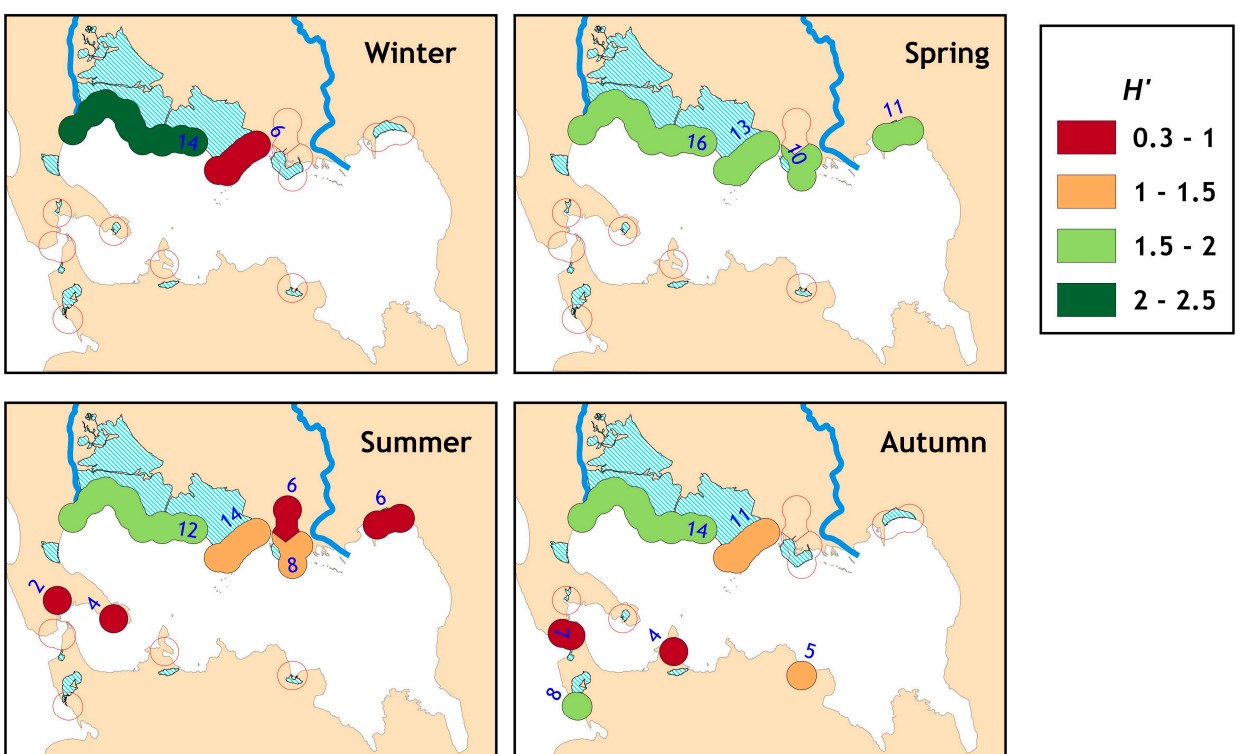

**Figure 2.** Spatio-temporal distribution of Shannon–Weiner index (H′) and number of the species (blue numbers) of all stages in the study area.

For the AS, high spatial distribution, non-seasonality, and no microhabitat selection (total score 222) were estimated for seven species/genera (*A. boyeri*, *Gobius* sp., *Palaemon* sp., *B. ocellaris*, *S. abaster*, *C. saliens*, and *A. fasciatus*). Low spatial distribution, non-seasonality, and no microhabitat selection (total score 122) were estimated for seven species/genera (*P. kerathurus*, shrimp, *S. typhle*, *C. aurata*, *C. ramada*, *S. aurata*, and *S. solea*). Low spatial distribution, seasonality, and no microhabitat selection (total score 112) were estimated for two species/genera (*B. belone* and *Serranus* sp.). Low spatial distribution, seasonality, and microhabitat selection (total score 111) were estimated for the 11 remaining species.

For the NR, high spatial distribution, non-seasonality, and no microhabitat selection (total score 222) were estimated for three species/genera (*A. boyeri*, *Gobius* sp., *Palaemon* sp.). Low spatial distribution, non-seasonality, and no microhabitat selection (total score 122) were estimated for six species/taxa (*B. ocellaris*, *S. abaster*, *C. saliens*, *P. kerathurus*, shrimp, and *S. typhle*). Low spatial distribution, seasonality, and no microhabitat selection (total score 112) were estimated for four species/genera (*A. fasciatus*, *C. aurata*, *C. ramada*, and *S. aurata*) and low spatial distribution, seasonality, and microhabitat selection (total score 111) were estimated for seven species/genera (*S. solea*, *B. ocellaris*, *Serranus* sp., *D. labrax*,

*G. holbrooki*, *M. cephalus*, and *Symphodus* sp.). For seven species/genera (*E. encrasicolus*, *Hippocampus* sp., *L. viridis*, *S. scrofa*, *T. pleurobipunctatus*, *S. pilchardus*, and *S. officinalis*), no NR individuals were recorded (Figure 3).

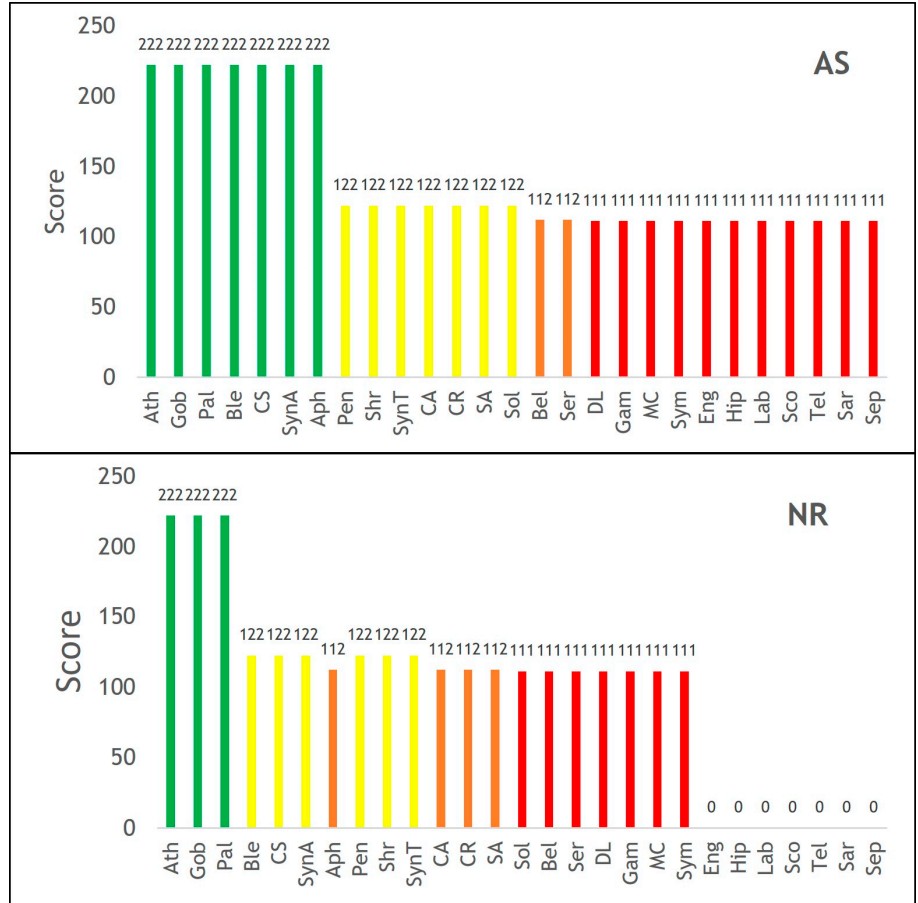

**Figure 3.** Ranking of species/genera according to their ecological profile for all ontogenetic stages (AS) and new recruits (NR): high spatial distribution, non-seasonality, and no microhabitat selection (score 222); high spatial distribution, non-seasonality, and microhabitat selection (total score 221); low spatial distribution, non-seasonality, and no microhabitat selection (total score 122); low spatial distribution, non-seasonality, and microhabitat selection (total score 121); low spatial distribution, seasonality, and no microhabitat selection (total score 112); and low spatial distribution, seasonality, and microhabitat selection (total score 111).

## 4. Discussion

Considering the significant role of Amvrakikos Gulf in fishing activities, the results of the present study support the fact that the coastal zone of the gulf serves as a nursery for numerous fish and decapods species/genera, which were the most representative species of the local fishery [5,7]. The average relative abundance of the species/genera exhibited no differences compared to other brackish waters of Greece (i.e., Porto Lagos Lagoon [14] and Drana Lagoon [15]), where the same sampling technique was used. The number of fish species/genera observed in the present study was less than that observed in other Greek brackish waters (present study: 21 species/genera; Porto Lagos Lagoon: 35 species [15]; Strymonikos systems: 43 species [16]). Given that *Gobius* sp., which in the present study exhibited an important presence (14.82% of total ind.), this species group includes at least five species (*Gobius cobitis*, *Gobius niger*, *Gobius paganellus*, *Pomatoschistus marmoratus*, and *Gobius ophiocephalus*) [16], thus increasing the number of species presented in this study up to twenty-six. It is worth noting that *G. ophiocephalus* represented 2% to 15% of the fishery catches in the Amvrakikos Gulf lagoons [7].

Individuals of all ontogenetic stages (fry, juveniles, and adults) were reported for several species (*A. boyeri*, *A. fasciatus*, *S. abaster*, *S. tyfle*, and *B. ocellaris*), indicating that they may be regarded as residents in the coastal zone, providing habitats for their entire life cycle. On the other hand, the number of species that appeared occasionally (up to two seasons) in the surrounding lagoons of the gulf was relatively less compared to other Greek brackish waters: 26 to 35 in Porto Lagos [14] and 27 to 45 in Strymonikos systems [16]. Eight species/genera were distributed to marine and brackish waters and exhibited occasional presence in the coastal zone of the gulf. Exceptions to this pattern were the *T. Pleurobipunctatus* and *G. holbrooki*, which were caught in freshwaters (SpC-11), and *S. scrofa*, which were caught in the coastal area (SpC-1), respectively, where the sampling took place only in one season (summer and autumn, respectively), and it could be falsely regarded as occasional species. The coastal zone of Amvrakikos Gulf consists of a transitional zone between its surrounding lagoons (brackish waters) and marine habitats, as demonstrated by the variations in the frequency of occasional species seen in the aforementioned research. This assumption is reinforced by the fact that the majority of the sampling stations were situated close to the lagoon and gulf communication inlets and that important commercial species of the coastal fishery [6] like *S. pichardus*, *S. officinalis*, and *M. barbatus* are either absent (red mullet) or only infrequently caught in the current study (*S. pichardus* and *S. officinalis*: collected from one individual).

Twenty to twenty-five species/genera exhibited new recruits, including those that spawn in the open sea (Ionian Sea) (Mugilidae, *S. aurata*, and *D. labrax* [9]) and those that spawn in the coastal zone (*A. boyeri*, *A. fasciatus*, and *Sygnathus* sp. [17–19]) or in the deeper zone (e.g., *P. kerathurus* [10,20]) of the gulf. Although *A. anguilla* and *C. labrosus* make up a considerable portion of the lagoon fishing catches [7], these species were not found in the samplings of this study. The absence of *A. anguilla*, which was caught by fyke nets [11], could be attributed to the different methods used for the samplings of the present study. On the other hand, *C. labrosus* may be found in the coastal zone for a brief length of time, reducing the likelihood of being caught during sampling.

The seasonal appearance of the new recruits of fish species in Amvrakikos Gulf was in agreement with the corresponding one estimated in other brackish waters in Greece [14–16], in the Mediterranean [21], and in the Atlantic [22]. In the Greek waters in particular, the seasonal appearance of new recruits in the present study is consistent with the seasonal appearance of the new recruits of the same species in the brackish waters of Western Greece (i.e., Messolonghi-Etolikon [23–25]) and of the north Aegean (i.e., Strymonikos Gulf [15–17]). The seasonal appearance of the new recruits of *M. kerathurus* is synchronized with and/or coincides with the species' spawning phase in the Gulf [10]. In contrast, *A. fasciatus* exhibited high abundance in muddy sand substrates with low plant cover and depth up to 0.5 m. This is in contrast to other studies in Greek brackish waters, as *A. fasciatus* prefers shallow waters and still/low-running water zones with abundant vegetation [26,27]. The small-sized individuals of *A. fasciatus* feed on planktonic prey (e.g., copepods, ostracods, and nauplii of Artemia), while larger sizes prefer larger and more benthic preys (e.g., amphipods and Bivalvia) [28]. *B. ocellaris*, *S. abaster*, and *S. typhle* exhibited high abundances in sandy and/or sandy gravel substrates with rich plant coverage, a pattern that is expected [29]. *Symphodus* sp., *L. viridis*, and *Hippocampus* sp. were present only in sandy and/or sandy gravel substrates with rich plant coverage, whereas *M. cephalus* and *D. labrax* were present only in muddy sand and sandy substrates with low or rich plant coverage. The fry of *D. labrax* feed on copepods and nauplii [30] and that of *M. cephalus* on copepods and algae [30], and the preference of these species in muddy sand substrates related to the availability of suitable food items in these microhabitats [31]. The presence of the invasive species *G. holbrooki* was recorded only in the station at freshwater regime (SpC-11), while the level of recorded salinities in some stations was within the range of species occurrence [32]. The presence of *G. holbrooki* in the SpC7 and SpC9 stations might be attributed to the low salinity of these stations. The salinity of the lagoon serves as a

barrier to the species in freshwater habitats [33], which exceeds the upper salinity limits of species occurrence.

In conclusion, the present study emphasizes the importance of the coastal zone as a nursery habitat for commercially important species. The species composition in the Amvrakikos Gulf at 10 cm and above was in agreement with the transitional nature of the area, with permanent and occasional species present. The latter includes species that migrate to lagoons on a regular basis as an optional aspect of their biological cycle [15,16]. Given the importance of Amvrakikos Gulf for local biodiversity and fishing activities, the present study presents certain practical implications for conservation. For instance, *S. abaster* is included in Appendix III—protected fauna species of the Bern Convention [34], and *A. fasciatus* is included in Annex II of the European Council Directive 92/43 [35].

**Supplementary Materials:** The following supporting information can be downloaded at https://www.mdpi.com/article/10.3390/d16030164/s1. Table S1: The species caught in the Amvrakikos Gulf coastal area and their relative abundance. Abr—abbreviation of the species name, INT: (F—fishing interest in lagoons, f—fishing interest in the gulf, P—protected species, INV—invasive species), L (SD)—total length (standard deviation), Range—total length range, n—number of individuals, %—relative frequency of individuals, LNR—separation total length new recruits and juvenile/adults, nNR—number of individuals new recruits, %NR—percentage of NR to total individuals of species, DAS and DNR—abundance of total individuals and new recruits, respectively, AS—total individuals of species, NR—new recruits of species, H′—Shannon–Weiner index, S—number of species. Table S2: Presence (dark for AS and grey for both NR and AS squares) and statistical higher values of abundance (+: for DAS, -: for DNR) (Kruskal–Wallis test and for post hoc Mann–Whitney U test) of species per spatial cluster (SpC), season (1: Jan–Mar; 2: Apr–Jun; 3: Jul–Sep; and 4: Oct–Dec) and microhabitat cluster (SC) in the Amvrakikos Gulf coastal area. AS—total individuals of species, NR—new recruits of species, DAS and DNR—abundance of total individuals and new recruits, respectively, Abr—abbreviation of species name (see Table S1), pr—number of SpC, season, or SC where the species is present (AS/NR), Sig—results of tests for statistically significant differences in abundance (*: $p < 0.05$; **: $p < 0.1$; ns—non-statistically significant difference; nt—not tested; nd—no data), H′—Shannon–Weiner index, S—number of species, AS pr—total number of species of category AS, NRpr—total number of species of category NR, nSamples—total number of samples. Figure S1: Spatial distribution of species/genera presence in the Amvrakikos Gulf.

**Author Contributions:** Conceptualization, G.K. and C.K.; methodology, G.K. and C.K.; resources, G.K. and N.V.; data curation, G.K., N.V. and D.K.M.; writing—review and editing, G.K., C.K. and D.K.M.; supervision, G.K.; project administration, C.K.; funding acquisition C.K. All authors have read and agreed to the published version of the manuscript.

**Funding:** Data collected through the project "Identification, consequences and management of the anoxic zone of Amvrakikos Gulf (NW Greece)" were funded by the European Economic Area Financial Mechanism (EEA FM) 2009–2014.

**Institutional Review Board Statement:** Not applicable.

**Data Availability Statement:** The original contributions presented in the study are included in the article and Supplementary Material, further inquiries can be directed to the first author.

**Acknowledgments:** The authors want to thank the fishers leasing the local lagoons for their assistance in the fish sampling.

**Conflicts of Interest:** The authors declare no conflicts of interest.

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
