# Peer review of "Diversity of Fish and Decapod Fry in the Coastal Zone of Amvrakikos Gulf"

_diversity, doi:10.3390/d16030164_

Round 1

Reviewer 1 Report

Comments and Suggestions for Authors

This manuscript provides lots of information on fish and decapod in the coastal zone of Amvrakikos Gulf, such as species composition, abundance, habitat preference, distribution characteristics, biodiversity, which forms the basis for protecting and managing the fish population and habitats. Some questions are provided for the authors consideration to improve the quality of this manuscript.

 1. The consistence among the title, the aim and the presented results

As least the aims given in Lines 13-14 and Lines 85-86 is not consistent with the title and the results. The readers may think that this paper focused on fry according to the aims. It seems that the results do not give the clear seasonal patterns of abundance according to the aims. In addition, “temporal” may not be suitable in the tile because the time span is not long. “species” in the title is not concrete.

I suggest that the authors refine the title and the aim according to the present results in the manuscript. Otherwise, the authors need to supplement results to demonstrate the findings on seasonal patterns of distribution, seasonal patterns of abundance for fry.

 2. Main results and conclusions need to be supplemented in the abstract. The current abstract mainly described the background and what the authors studied. Main results and conclusions were not summarized. As a result, readers cannot obtain the key information that this paper will provide by reading the abstract.

 3. the discussion section

The authors are suggested to add discussions on main findings. For instance, according to the title, the main findings are on spatial distribution of diversity, and seasonal distribution of diversity. While according to the aims, the main findings are on seasonal patterns of distribution of fry, and seasonal patterns of abundance of fry. The current discussions are not focused on main findings.

 4. SC3 should be SC4 in Line 177.

Author Response

We would like to thank Reviewer 1 for his/her fruitful comments and suggestions that highly improve the quality and chariness of our manuscript. In general, the text has been revised by an English native speaker and we reply to his/her comments accordingly. Also, changes are pointed with track changes within the text.

Reviewer 2 Report

Comments and Suggestions for Authors

The abstact does not report the results of the statistical survey. Add a summary of the main results for the spatial and temporal distribution of species

Modify the following words/phrases:

line 110           change "Season"  to "season" 

lines 108-114  specify the software/s used

line 172           change   "Table I"  to  "table 1"

lines 190-191  move the sentence     "*1: 190 Mud, 2: Muddy Sand, 3: Sand, 4: Sandy Gravel"  below the table

Table 1    changing the arrengment of the word   "num-ber/SpC"

Table 2 is not shown in the text

Author Response

We would like to thank Reviewer 2 for his/her fruitful comments and suggestions that highly improve the quality and chariness of our manuscript. In general, the text has been revised by an English native speaker. Changes are pointed with track changes within the text.

Reviewer 3 Report

Comments and Suggestions for Authors

The manuscript by George Katselis and co-authors represents interesting information about inhabitants of important Greek brackish waters in Amvrakikos Gulf. Design of the experiment seems to be fine. Reasonable statistics was used to evaluate differences between subsets and to highlight the main findings.

Major comments:

I suggest adding the region to the title of the manuscript e.g., ‘: Amvrakikos Gulf, W Greece’.

Abstract is too descriptive and does not provide real findings. I suggest adding additional information from Results and Discussion.

The list of keywords is too poor.

Please, add the aim of the research and scientific hypothesis checked.

Please, replace ‘season no’ to the name of season.

Results section is too descriptive. I suggest to make new resulting figures and tables (look at my comments below) and shorten the text.

Specific comments:

L.25 – 31. Please, shorten.

L. 39. “i.e. Arachthos and Louros”. Gulfs or rivers?

L. 71 – 73. Please, shorten or correct the phrase.

L. 115 – 122. I suggest calculating indVal (Multilevel pattern analysis) in an «indicspecies» package for R [De Cáceres, Legendre, 2009]. Please, consider.

L.135 – 170. Please, shorten as it is too descriptive. I suggest adding figure(s).

L. 152 ‘137.3±103.6’, hereinafter. Please, use integer.

L. 171 – 180 repeats Table 1. Please, shorten.

L. 181 – 186. Please, provide table or a plot with box and whickers for mentioned 10 spatial clusters.

L. 193 – 204, 213 – 219. Please, convert this information in Table to increase readability.

L. 219 – 226. Please, shorten.

L. 231 ‘six species/genus’ and L. 233 ‘seven species/genus’. Please, change.

L. 241 ‘six species/genus’. Five?

L. 256 ‘five species/genus’. Seven?

L. 230 – 247 and figure 3. Please, shorten and convert to table to increase readability.

Author Response

We would like to thank Reviewer 3 for his/her fruitful comments and suggestions that highly improve the quality and chariness of our manuscript. In general, the text has been revised by an English native speaker.
